

# SSMFN: a fused spatial and sequential deep learning model for methylation site prediction

Favorisen Rosyking Lumbanraja[1,*], Bharuno Mahesworo[2,3,*], Tjeng Wawan Cenggoro[2,4], Digdo Sudigyo[2] and Bens Pardamean[2,5]

[1] Department of Computer Science, Faculty of Mathematics and Natural Science, University of Lampung, Bandar Lampung, Lampung, Indonesia
[2] Bioinformatics and Data Science Research Center, Bina Nusantara University, West Jakarta, Jakarta, Indonesia
[3] Statistics Departement, School of Computer Science, Bina Nusantara University, West Jakarta, Jakarta, Indonesia
[4] Computer Science Departement, School of Computer Science, Bina Nusantara University, West Jakarta, Jakarta, Indonesia
[5] Computer Science Department, BINUS Graduate Program - Master of Computer Science, Bina Nusantara University, West Jakarta, Jakarta, Indonesia
* These authors contributed equally to this work.

Corresponding author
Bharuno Mahesworo,
bharuno.mahesworo@binus.edu

## ABSTRACT

**Background**. Conventional *in vivo* methods for post-translational modification site prediction such as spectrophotometry, Western blotting, and chromatin immune precipitation can be very expensive and time-consuming. Neural networks (NN) are one of the computational approaches that can predict effectively the post-translational modification site. We developed a neural network model, namely the Sequential and Spatial Methylation Fusion Network (SSMFN), to predict possible methylation sites on protein sequences.

**Method**. We designed our model to be able to extract spatial and sequential information from amino acid sequences. Convolutional neural networks (CNN) is applied to harness spatial information, while long short-term memory (LSTM) is applied for sequential data. The latent representation of the CNN and LSTM branch are then fused. Afterwards, we compared the performance of our proposed model to the state-of-the-art methylation site prediction models on the balanced and imbalanced dataset.

**Results**. Our model appeared to be better in almost all measurement when trained on the balanced training dataset. On the imbalanced training dataset, all of the models gave better performance since they are trained on more data. In several metrics, our model also surpasses the PRMePred model, which requires a laborious effort for feature extraction and selection.

**Conclusion**. Our models achieved the best performance across different environments in almost all measurements. Also, our result suggests that the NN model trained on a balanced training dataset and tested on an imbalanced dataset will offer high specificity and low sensitivity. Thus, the NN model for methylation site prediction should be trained on an imbalanced dataset. Since in the actual application, there are far more negative samples than positive samples.

# INTRODUCTION

Methylation is a post-translational modification (PTM) process that modifies the functional and conformational changes of a protein. The addition of a methyl group to the protein structure plays a role in the epigenetic process, especially in histones (*Lee et al., 2005*). Histone methylation in Arginine (R) and Lysine (K) residues substantially affects the level of gene expression along with other PTM processes such as acetylation and phosphorylation (*Schubert, Blumenthal & Cheng, 2006*). Moreover, methylation directly alters the regulation, transcription, and structure of chromatin (*Bedford & Richard, 2005*). Genetic alterations through the methylation process induce oncogenes and tumor suppressor genes that play a crucial role in carcinogenesis and metastasis cancer (*Zhang et al., 2019*).

Currently, most of the methods for PTM sites prediction were conducted by implementing *in vivo* methods, such as Mass Spectrophotometry, Western Blotting, and Chromatin Immune Precipitation (ChIP). However, computational (in silico) approaches are starting to be more popular for PTM sites prediction, especially methylation. Computational approaches for predicting protein methylation sites can be an inexpensive, highly accurate, and fast alternative method through massive data sets. The commonly used computational approaches are support vector machine (SVM) (*Chen et al., 2006*; *Shao et al., 2009*; *Shien et al., 2009*; *Shi et al., 2012*; *Lee et al., 2014*; *Qiu et al., 2014*; *Wen et al., 2016*), group-based prediction system (GPS) (*Deng et al., 2017*), Random Forest (*Wei et al., 2017*), and neural network (NN) (*Chen et al., 2018*; *Hasan & Khatun, 2018*; *Chaudhari et al., 2020*).

The application of the machine learning approach to predict possible methylation sites on protein sequences has been studied in numerous previous research. The latest and the most relevant studies to our study were conducted by *Chen et al. (2018)* and *Chaudhari et al. (2020)*. *Chen et al. (2018)* developed MUscADEL (Multiple Scalable Accurate Deep Learner for lysine PTMs), a methylation site prediction model that was trained and tested on human and mice protein data sets. MUscADEL utilized bidirectional long short term memory (LSTM) (*Graves & Schmidhuber, 2005*). Meanwhile, *Chen et al. (2018)* hypothesized that the order of amino acids in the protein sequence has a significant influence on the location where the methylation process can occur. The other model is DeepRMethylSite which was developed by *Chaudhari et al. (2020)*. The model was implemented with the combination of convolutional neural network (CNN) and LSTM. The combination was expected to be able to extract the spatial and sequential information of the amino acids sequences.

Before the practical application by *Chaudhari et al. (2020)* to predict methylation site, a combination of LSTM and CNN approach has been implemented since 2015 by *Xu, Li & Deng (2015)* to strengthen a face recognition model. This combination was also found In the natural language processing (NLP) area. For instance, *Wang et al. (2016)* developed a dimensional sentiment analysis model and suggested that a combination of LSTM and CNN is capable of capturing long-distance dependency and local information patterns. Related to NLP, *Wu et al. (2018)* developed an LSTM-CNN model with similar architecture to other previous studies where the CNN layer and LSTM layer were implemented in a serial

structure. Recently, the combination of CNN and LSTM was also applied for educational data (*Prabowo et al., 2021*).

In this study, we developed the Sequential and Spatial Methylation Fusion Network (SSMFN) to predict possible methylation sites on the protein sequence. Similar to DeepRMethylSite, SSMFN also utilized CNN and LSTM. However, instead of treating them as an ensemble model, we fused the latent representation of the CNN and LSTM modules. By allowing more relaxed interaction between the CNN and LSTM modules, we hypothesized that the fusion approach can extract better features than the model with the ensemble approach.

## METHODS

### Dataset

The dataset in this study was obtained from the previous methylation site prediction study by *Kumar et al. (2017)*. The data was collected from other studies as well as from Uniprot protein database (*Apweiler et al., 2004*). The collected data was furthermore experimentally verified *in vivo*.

The dataset comprises sequences of 19 amino acids with arginine in the middle of the sequence because the possible location for methylation is on arginine (R). These sequences are segments from the full amino acids sequence. Examples of the amino acids sequences in this dataset are shown in Table 1. The dataset was split into three datasets: training, validation, and independent dataset. Each dataset contains positive and negative samples, where positive samples are the sequence where methylation occurs in the middle amino acid. The distribution of each dataset can be seen in Table 2. Because the original dataset was imbalanced, previous studies often constructed a new balanced dataset to improve the performance of their model. This practice is needed because most machine learning methods are not robust to imbalanced training data. Following the typical practice in previous studies, we also created a balanced training dataset as well as a balanced validation dataset for a fair comparison.

### Experiment

First, to understand the contribution of each element in the proposed model, we carried an ablation study on our proposed model. The elements tested and explored in this ablation study were the CNN and LSTM branches of the model. Afterward, we compared the performance of our proposed model to DeepRMethylSite (*Chaudhari et al., 2020*). Additionally, we also provided a comparison to a standard multi-layer perceptron model. To measure the effect of the data distribution (balanced or imbalanced), we conducted separate experiments for the balanced and the original imbalanced dataset. Afterward, the trained models from both experiments were validated and tested on the balanced validation dataset, the imbalanced validation dataset, and the test dataset, respectively. The workflow of this study is illustrated in Fig. 1. All models in the experiment were developed using Python machine learning library, PyTorch (*Paszke et al., 2019*). To train the models, we utilized a NVIDIA Tesla P100 Graphical Processing Unit (GPU) as well as a publicly available GPU instance provided by Google Colab.

**Table 1  Protein sequence dataset example.**

| No | Sequence | | | | | | | | | | | | | | |
|---|---|---|---|---|---|---|---|---|---|---|---|---|---|---|---|
| | 1st | 2nd | 3rd | . | . | 8th | 9th | 10th | 11th | 12th | . | . | 17th | 18th | 19th |
| 1 | V | E | S | . | . | V | T | **R** | L | H | . | . | H | M | N |
| 2 | K | N | H | . | . | I | S | **R** | H | H | . | . | D | P | Q |
| 3 | H | P | P | . | . | R | L | **R** | G | I | . | . | W | D | H |
| . | . | . | . | . | . | . | . | . | . | . | . | . | . | . | . |
| . | . | . | . | . | . | . | . | . | . | . | . | . | . | . | . |
| n | R | S | I | . | . | A | C | **R** | I | R | . | . | K | W | Y |

**Table 2  Amino acids sequences dataset list.**

| Data class | Label | *n* sequences |
|---|---|---|
| *Training* | Positive | 1,038 |
| | Negative | 5,190 |
| *Balanced training* | Positive | 1,038 |
| | Negative | 1,038 |
| *Validation* | Positive | 1,131 |
| | Negative | 3,033 |
| *Balanced validation* | Positive | 1,131 |
| | Negative | 1,131 |
| *Independent* (*Test*) | Positive | 260 |
| | Negative | 260 |

### Spatial and sequential methylation fusion network (SSMFN)

Our proposed model, the Spatial and Sequential Methylation Fusion Network (SSMFN), was designed with the motivation that a protein sequence can be perceived as both spatial and sequential data. The view of a protein sequence as spatial data assumes that the amino acids are arranged in a one-dimensional space. On the other hand, protein sequences can also be thought of as sequential data by assuming that the next amino acid is the next time step of particular amino acid. On modelling protein sequences with deep learning, CNN is applied when adopting spatial data view, while LSTM is applied for the sequential data. Using the information from both views has been shown to be beneficial by *Chaudhari et al. (2020)*. Their model was implemented by having an ensemble model of CNN and LSTM that read the same sequence. However, *Chaudhari et al. (2020)* processed the spatial and sequential view with separate sub-models. As a consequence, it cannot extract joint spatial-sequential features, which might be beneficial in modelling protein sequences. Having observed that, we constructed SSMFN as a deep learning model with an architecture that can fuse the latent representation of CNN modules and LSTM modules.

To read the amino acid sequence, SSMFN applied an embedding layer with 21 neurons. This embedding layer was used to enhance the expression of each amino acid. Thus, the number of neurons in this layer matches the amounts of amino acids variants. Therefore, each type of amino acid can have a different vector representation. The output of this layer

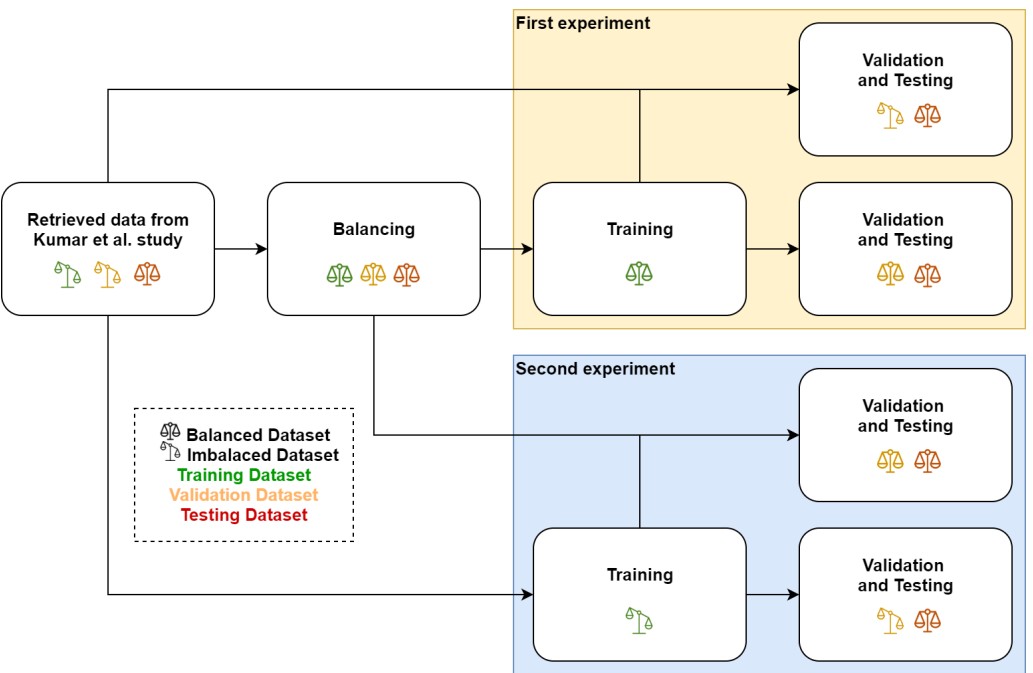

**Figure 1 Research workflow.** The chart shows that the data we used in this research was retrieved from *Kumar et al. (2017)*. The data was afterward balanced accordingly. In the first experiment, we trained our model using the balanced training dataset. Subsequently, we validated and tested the model on the balanced and the imbalanced dataset. We did a similar workflow for the second experiment. However, instead of the balanced dataset, we trained the model on the imbalanced training dataset.

is then split into LSTM and CNN branches. In the LSTM branch, we created two LSTM layers with 64 neurons each. Every LSTM layer is followed by a dropout layer with a 0.5 drop rate. It is subsequently followed by a fully connected layer at the end of the branch with 32 neurons. This fully connected layer serves as a latent representation generator that is fused with the latent representation from the CNN branch.

In contrast, the CNN branch comprised four CNN layers with 64 neurons in each layer. Unlike the LSTM layers, residual connections were utilized in the CNN branch. Each CNN layer is a 2D convolutional layer with rectified linear units (ReLU) as the activation function. Every CNN layer also has a 2D batch normalization layer and a dropout layer which is set at 0.5. At the end of the branch, a fully connected layer with 32 neurons is installed to match the output with the LSTM branch.

In the next step, the latent representation of both branches was fused with a summation operation. The fused representation was subsequently processed through a fully connected layer with two neurons as the last layer. This layer predicts whether the methylation occurred at the center of the amino acid or not. The architecture of the proposed model and the hyperparameter settings is illustrated in Fig. 2 and listed in Table 3. The code of this model can be accessed in the following link: https://github.com/bharuno/SSMFN-Methylation-Analysis.

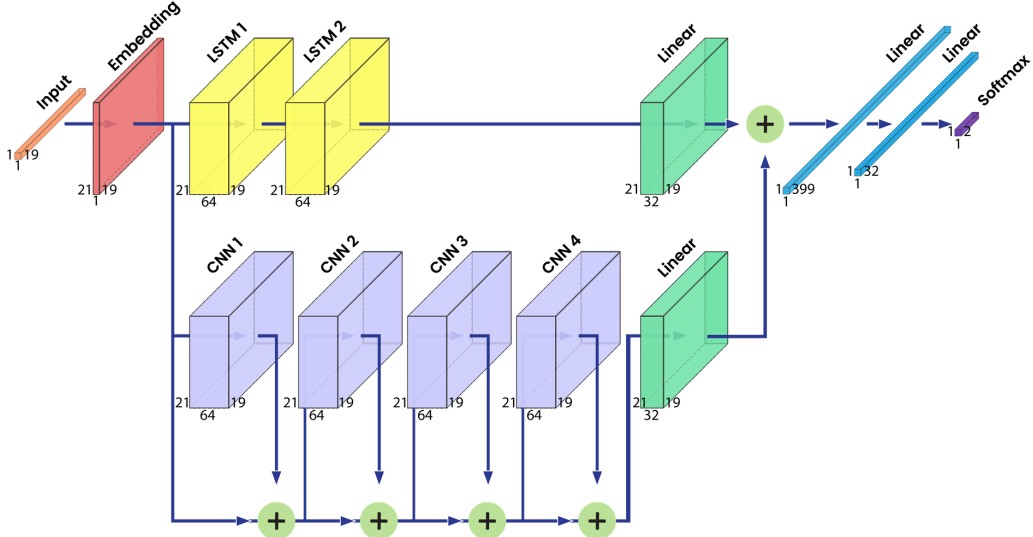

**Figure 2** Proposed neural network architecture.

**Table 3** Hyperparameter settings.

| Parameter | Settings |
|---|---|
| Learning rate | 0.001 |
| Epochs | 500 |
| Optimizer | Adam |
| Embedding layer neurons | 21 |
| Embedding layer output dimension | $21 \times 19 = 399$ |
| Output layer neurons | 2 |
| **LSTM Branch** | |
| LSTM layers neurons | 64 |
| Dropout layers drop rate | 0.5 |
| Fully connected layer neurons | 32 |
| **CNN Branch** | |
| CNN layers neurons | 64 |
| CNN layers activation function | Rectified linear units |
| Dropout layers drop rate | 0.5 |
| Fully connected layer neurons | 32 |

### Comparison to a standard multi-layer perceptron

A standard multi-layer perceptron (SMLP) NN was developed to be compared to our proposed model. This multi-layer perceptron model was included in this study to provide an insight into the performance of a simple model to solve the methylation site prediction problem. This model consists of an embedding layer followed by two fully connected layers. The embedding layer has 21 neurons because there are 21 types of amino acids. The first fully connected layer has 399 neurons which came from 21 (types of amino acid) multiplied by 19 (protein sequence length). After the first layer, we put a second fully connected layer

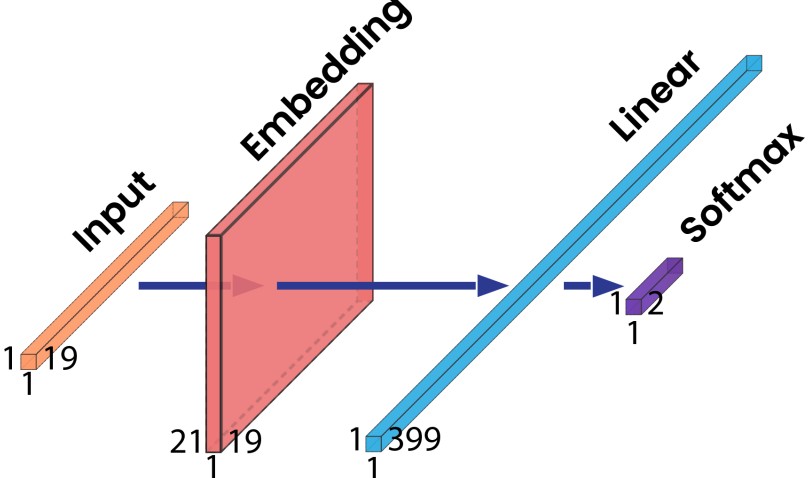

**Figure 3  The standard multi-layer perceptron architecture.**

that has two neurons as the output for prediction. The structure of this model is shown in Fig. 3.

### Comparison to DeepRMethylSite

For a fair comparison of our proposed model to other state-of-the-art methylation site prediction models, we re-conducted the experiment to train DeepRMethylSite (*Chaudhari et al., 2020*) with the same dataset used by our proposed model. To obtain optimal DeepRMethylSite performance on our dataset, we adjusted several hyperparameters. First, we changed the LSTM branch optimizer, from Adadelta to Adam. Second, we removed recurrent dropout layers in the LSTM branch. Finally, we set the maximum number of epochs to 500.

### Evaluation

To evaluate the performance of the proposed model and to compare it to the models from previous studies, we utilized Accuracy (Eq. (1)), Sensitivity (Eq. (2)), Specificity (Eq. (3)), F1 score (Eq. (4)), Matthews correlation coefficient (MCC) (Eq. (5)), and area under curve (AUC) (*Bradley, 1997*). These metrics were commonly employed in the previous research with a focus on prediction protein phosphorylation site (*Lumbanraja et al., 2018*; *Lumbanraja et al., 2019*). The AUC was computed using the scikit-learn library from the receiver operating characteristic (ROC) of the models' performance.

$$Accuracy = \frac{TP + TN}{TP + TN + FP + FN} \tag{1}$$

$$Sensitivity = \frac{TP}{TP + FN} \tag{2}$$

$$Specificity = \frac{TN}{TN + FP} \tag{3}$$

$$F1score = \frac{TP}{TP + FP + FN} \tag{4}$$

$$MCC = \frac{(TP * TN) - (FP * FN)}{\sqrt{(TP + FP)(TP + FN)(TN + FP)(TN + FN)}} \tag{5}$$

## RESULTS

Tables 4 and 5 show the results obtained from our ablation study. Meanwhile, Tables 6 and 7 summarized the comparative results of our model to the previous models with the balanced and imbalanced training dataset, respectively. In Table 6, we also added the performance of several methylation site prediction models from previous studies including MeMo (*Chen et al., 2006*), MASA (*Shien et al., 2009*), BPB-PPMS (*Shao et al., 2009*), PMeS (*Shi et al., 2012*), iMethylPseAAC (*Qiu et al., 2014*), PSSMe (*Wen et al., 2016*), MePred-RF (*Wei et al., 2017*) and PRmePRed (*Kumar et al., 2017*). The performances of MeMo, MASA, BPB-PPMS, PMeS, iMethylPseAAC, PSSMe and MePred-RF were reported by *Chaudhari et al. (2020)*. Meanwhile, the performance of PRmePRed was reported by *Kumar et al. (2017)*.

## DISCUSSION

The results of the ablation study in Tables 4 and 5 show that the LSTM branch and CNN branch achieved better performance compared to the merged model at least on one dataset. However, the merged models achieved better performance in most of the datasets, specifically in the test dataset. This fact indicates that the merged model has a better generalization capability than the model with only CNN or LSTM branches.

In the experiment on the balanced training dataset, our proposed model emerged as the best NN model with the best performance in all metrics except sensitivity among all other NN models. Interestingly, the DeepRMethylSite final result (merged) was not better in all metrics compared to its CNN branch and its LSTM branch. On the imbalanced validation dataset, our proposed model, SSMFN, has more than 4% higher accuracy and 6% higher MCC which is the best parameter for assessing model performance on imbalanced data, compared to the DeepRMethylSite model. On the balanced validation dataset and test dataset, SSMFN has 2–4% higher accuracy compared to DeepRMethylSite.

In Table 6, we also present the performance of other methylation site prediction models from previous studies as reported by *Chen et al. (2018)* and *Chaudhari et al. (2020)*. The models from previous studies provided an overview of the performance of non-neural-network models. The best non-neural-network model, PRmePRed, has more than 5% higher accuracy than SSMFN. However, it should be noticed that non-neural-network models require heavy feature engineering, which is also found in PRmePRed. This introduced unnecessary manual labor that can be avoided by the utilization of modern NN models, which are also known as deep learning. Interestingly, the SMLP

**Table 4** The first ablation study, trained on the balanced training dataset.

| Model | Acc | F1 | Sens | Spec | MCC | AUC |
|---|---|---|---|---|---|---|
| **Validated on the imbalanced validation dataset** | | | | | | |
| SSMFN CNN | 0.7891 | 0.7649 | 0.5745 | 0.9368 | 0.5649 | 0.8120 |
| SSMFN LSTM | **0.8252** | **0.7985** | **0.6328** | 0.9354 | **0.6148** | 0.8326 |
| SSMFN Merged | 0.8187 | 0.7943 | 0.6175 | **0.9442** | 0.6143 | **0.8359** |
| **Validated on the balanced validation dataset** | | | | | | |
| SSMFN CNN | **0.8431** | **0.8427** | **0.8767** | 0.8149 | **0.6889** | 0.8120 |
| SSMFN LSTM | 0.8302 | 0.3020 | 0.8195 | 0.8417 | 0.6609 | 0.8326 |
| SSMFN Merged | 0.8360 | 0.8358 | 0.8130 | **0.8626** | 0.6738 | **0.8359** |
| **Tested on the test dataset** | | | | | | |
| SSMFN CNN | 0.7962 | 0.7960 | **0.8105** | 0.7831 | 0.5929 | 0.7962 |
| SSMFN LSTM | 0.7981 | 0.7980 | 0.8063 | 0.7903 | 0.5964 | 0.7981 |
| SSMFN Merged | **0.8115** | **0.8115** | 0.8000 | **0.8240** | **0.6235** | **0.8115** |

Note.
The highest value of each parameter from each measurement experiment is shown in bold.

**Table 5** The second ablation study, trained on the imbalanced training dataset.

| Model | Acc | F1 | Sens | Spec | MCC | AUC |
|---|---|---|---|---|---|---|
| **Validated on the imbalanced validation dataset** | | | | | | |
| SSMFN CNN | 0.8939 | 0.8502 | **0.9389** | 0.8834 | 0.7230 | 0.8179 |
| SSMFN LSTM | **0.9167** | **0.8891** | 0.9100 | **0.9186** | **0.7836** | **0.8704** |
| SSMFN Merged | 0.9078 | 0.8774 | 0.8895 | 0.9133 | 0.7598 | 0.8596 |
| **Validated on the balanced validation dataset** | | | | | | |
| SSMFN CNN | 0.7529 | 0.7372 | **0.9948** | 0.6698 | 0.5798 | 0.8179 |
| SSMFN LSTM | 0.8638 | 0.8624 | 0.9567 | **0.8024** | **0.7560** | **0.8704** |
| SSMFN Merged | **0.8656** | **0.8640** | 0.9672 | 0.8003 | 0.7491 | 0.8596 |
| **Tested on the test dataset** | | | | | | |
| SSMFN CNN | 0.7404 | 0.7228 | **0.9845** | 0.6598 | 0.5566 | 0.7404 |
| SSMFN LSTM | 0.8442 | 0.8418 | 0.9590 | **0.7754** | 0.7110 | 0.8442 |
| SSMFN Merged | **0.8462** | **0.8435** | 0.9688 | 0.7744 | **0.7173** | **0.8462** |

Note.
The highest value of each parameter from each measurement experiment is shown in bold.

model provided slightly better performance than DeepRMethylSite on the test dataset. This does not implicate that the SMLP model has a better performance compared to the DeepRMethylSite it has relatively poor performance in the validation dataset, both balanced and imbalanced.

When trained on the balanced training dataset and tested on the imbalanced validation dataset, most of the models have high specificity and low sensitivity. This phenomenon is normal since the training and test dataset have different distributions. Because the distribution of methylation is naturally imbalanced, this result suggested that we need to train methylation site prediction models on a dataset with its natural distribution for a practical purpose, not a balanced dataset.

**Table 6  The first experiment, trained on the balanced training dataset.**

| Model | Acc | F1 | Sens | Spec | MCC | AUC |
|---|---|---|---|---|---|---|
| **Validated on the imbalanced validation dataset** | | | | | | |
| DeepRMethylSite CNN | 0.7819 | 0.7557 | 0.5668 | 0.9259 | 0.5428 | 0.7990 |
| DeepRMethylSite LSTM | 0.7699 | 0.7479 | 0.5480 | 0.9394 | 0.5430 | 0.8024 |
| DeepRMethylSite Merged | 0.7743 | 0.7518 | 0.5474 | 0.9394 | 0.5481 | 0.8021 |
| SMLP | 0.7209 | 0.7018 | 0.4922 | 0.9281 | 0.4719 | 0.7649 |
| SSMFN Merged | **0.8187** | **0.7943** | **0.6175** | **0.9442** | **0.6143** | **0.8359** |
| **Validated on the balanced validation dataset** | | | | | | |
| DeepRMethylSite CNN | 0.8090 | 0.8089 | 0.7944 | 0.8251 | 0.6188 | 0.7990 |
| DeepRMethylSite LSTM | 0.7993 | 0.7993 | 0.7618 | 0.8493 | 0.6048 | 0.8024 |
| DeepRMethylSite Merged | 0.8059 | 0.8051 | 0.7659 | 0.8504 | 0.6169 | 0.8021 |
| SMLP | 0.7073 | 0.7073 | 0.7041 | 0.7107 | 0.4147 | 0.7649 |
| SSMFN Merged | **0.8360** | **0.8358** | **0.8130** | **0.8626** | **0.6738** | **0.8359** |
| **Tested on the test dataset** | | | | | | |
| MeMo* | 0.68 | na | 0.38 | 0.99 | 0.46 | na |
| MASA* | 0.65 | na | 0.31 | 0.99 | 0.41 | na |
| BPB-PPMS* | 0.56 | na | 0.12 | **1.00** | 0.25 | na |
| PMeS* | 0.58 | na | 0.43 | 0.73 | 0.16 | na |
| iMethyl-PseAAC* | 0.59 | na | 0.18 | **1.00** | 0.3 | na |
| PSSMe* | 0.72 | na | 0.6 | 0.83 | 0.44 | na |
| MePred-RF* | 0.69 | na | 0.41 | 0.97 | 0.46 | na |
| PRmePRed** | **0.8683** | na | **0.8709** | 0.8660 | **0.7370** | **0.9000** |
| DeepRMethylSite CNN | 0.7846 | 0.7846 | 0.7803 | 0.7891 | 0.5693 | 0.7846 |
| DeepRMethylSite LSTM | 0.8000 | 0.7989 | 0.7617 | 0.8514 | 0.6065 | 0.8000 |
| DeepRMethylSite Merged | 0.7942 | 0.7929 | 0.7508 | 0.8447 | 0.5959 | 0.7904 |
| SMLP | 0.8077 | 0.8076 | 0.8175 | 0.7985 | 0.6157 | 0.8077 |
| SSMFN Merged | 0.8115 | **0.8115** | 0.8000 | 0.8240 | 0.6235 | 0.8115 |

**Note.**
The highest value of each parameter from each measurement experiment is shown in bold.

In the second experiment, we trained the models using the imbalanced dataset with a 5 to 1 ratio for negative to positive size samples, respectively. Overall, our model achieved better performance when trained on the imbalanced dataset compared to the balanced dataset. Trained on the imbalanced dataset, SSMFN can even outperform PRmePRed in several metrics. SSMFN accuracy is 0.36% lower than the DeepRMethylSite accuracy on the imbalanced validation dataset. However, it has better performance on the balanced validation dataset and the test dataset compared to DeepRMethylSite.

## CONCLUSIONS

In general, our proposed model, SSMFN, provided better performance compared to DeepRMethylSite. Our model also performed better when trained on the imbalanced training dataset that it even has better performance than the model that uses feature extraction in several metrics. Additionally, we observed that all the NN models, including ours, achieved a high specificity and a low sensitivity when they were trained on the balanced

**Table 7  Second experiment, trained on the imbalanced training dataset.**

| Model | Acc | F1 | Sens | Spec | MCC | AUC |
|---|---|---|---|---|---|---|
| **Validated on the imbalanced validation dataset** | | | | | | |
| DeepRMethylSite CNN | 0.8948 | 0.8550 | 0.9072 | 0.8916 | 0.7242 | 0.8283 |
| DeepRMethylSite LSTM | 0.9092 | 0.8782 | 0.9044 | 0.9106 | 0.7634 | 0.8576 |
| DeepRMethylSite Merged | **0.9114** | **0.8808** | 0.9047 | 0.9115 | **0.7693** | 0.8589 |
| SMLP | 0.9071 | 0.8670 | **0.9973** | 0.8873 | 0.7635 | 0.8295 |
| SSMFN Merged | 0.9078 | 0.8774 | 0.8895 | **0.9133** | 0.7598 | **0.8596** |
| **Validated on the balanced validation dataset** | | | | | | |
| DeepRMethylSite CNN | 0.8289 | 0.8249 | 0.9709 | 0.7527 | 0.6899 | 0.8283 |
| DeepRMethylSite LSTM | 0.8576 | 0.8557 | 0.9644 | 0.7908 | 0.7350 | 0.8576 |
| DeepRMethylSite Merged | 0.8585 | 0.8567 | 0.9645 | 0.7919 | 0.7365 | 0.8589 |
| SMLP | 0.7582 | 0.7432 | **1.0000** | 0.6740 | 0.5899 | 0.8295 |
| SSMFN Merged | **0.8656** | **0.8640** | 0.9672 | **0.8003** | **0.7491** | **0.8596** |
| **Tested on the test dataset** | | | | | | |
| DeepRMethylSite CNN | 0.7808 | 0.7727 | 0.9506 | 0.7039 | 0.6063 | 0.7808 |
| DeepRMethylSite LSTM | 0.8115 | 0.8070 | 0.9500 | 0.7382 | 0.6548 | 0.8115 |
| DeepRMethylSite Merged | 0.8135 | 0.8088 | 0.9553 | 0.7390 | 0.6598 | 0.8135 |
| SMLP | 0.7250 | 0.7025 | **1.0000** | 0.6452 | 0.5388 | 0.7250 |
| SSMFN Merged | **0.8462** | **0.8435** | 0.9688 | **0.7744** | **0.7173** | **0.8462** |

Note.
The highest value of each parameter from each measurement experiment is shown in bold.

dataset and tested on the imbalanced dataset. This suggested that, in future works, we need to consider using a dataset with the original distribution for training. This will train the models to recognize the real distribution of the methylation site prediction task, which has far more negative than positive samples, leading to better performance in practice.

## ACKNOWLEDGEMENTS

This research is a collaboration between Bioinformatics and Data Science Research Center (BDSRC), Bina Nusantara University and Department of Computer Science, Faculty of Mathematics and Natural Science, University of Lampung. The GPU Tesla P100 used to conduct the experiment was provided by NVIDIA - BINUS AIRDC.

### Funding
The authors received no funding for this work.

### Competing Interests
The authors declare there are no competing interests.

## Author Contributions

- Favorisen Rosyking Lumbanraja conceived and designed the experiments, authored or reviewed drafts of the paper, supervised and guided the research, and approved the final draft.
- Bharuno Mahesworo performed the experiments, analyzed the data, performed the computation work, prepared figures and/or tables, authored or reviewed drafts of the paper, and approved the final draft.
- Tjeng Wawan Cenggoro conceived and designed the experiments, prepared figures and/or tables, and approved the final draft.
- Digdo Sudigyo analyzed the data, prepared figures and/or tables, literature review, and approved the final draft.
- Bens Pardamean analyzed the data, authored or reviewed drafts of the paper, supervised and guided the research, and approved the final draft.

## Data Availability

The dataset and code are available at GitHub and in the Supplemental Files: https://github.com/bharuno/SSMFN-Methylation-Analysis, and https://github.com/bharuno/SSMFN-Methylation-Analysis/blob/main/SSMFN_dev.ipynb.

## Supplemental Information

Supplemental information for this article can be found online at http://dx.doi.org/10.7717/peerj-cs.683#supplemental-information.

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
