# Peer review of "SSMFN: a fused spatial and sequential deep learning model for methylation site prediction"

_PeerJ Computer Science, doi:10.7717/peerj-cs.683_

## Round 0.1 · original submission · Major Revisions

Comparative experiments are not sufficient. Please give more results compared with other methods on more datasets.

Reviewer 1 ·

Basic reporting

1) Please attention to some word mistakes, such as “swift and effectively” and “performers” in the abstract. Please check the words and grammar of the whole paper again.
2) The Fig. 3 can be more professional and informative drawn by specialized tools, such as PlotNeuralNet.

Experimental design

Comparative experiments are not sufficient. Please give more results compared with other methods on more datasets.

Validity of the findings

1)I think that fusing CNN and LSTM branch with a weighted operation is more rational. Please give reasons for simple summation.
2)DeepRMethylSite uses one hot encoding, where each amino acid is defined as a 20 length vector, with only one of the 20 bits as 1. And SSMFN uses an embedding layer with 21 neurons to encode different amino acids. What is the difference of those two representations and advantages of SSMFN?

Additional comments

This paper proposes a neural network model which combines CNN and LSTM to extract spatial and sequential information from amino acid sequences, respectively. The writing structure is reasonable, but there are some problems. Please see the detailed comments above.

Reviewer 2 ·

Basic reporting

no comment

Experimental design

no comment

Validity of the findings

no comment

Additional comments

1. We note that your model achieves good results, but the explanation of the structure of the model is not sufficient, and we hope it can be improved.

2. For Figure2, we would like you to have more visual representations and less textual representations.

3. You can try to center the content in Table1 to make it look more beautiful.

4. The literature review of the manuscript need to be improved by comparing similar papers. Thus, the readers can compare the results of various works and the main novelty of this paper can be easily identified.

---

## Round 0.2 · Minor Revisions

This is a well written manuscript, the authors tried to use deep learning method solve the problem, the proposed deep network architecture seems working well on the prediction, achieved a better accuracy than peer method.

Reviewer 1 ·

Basic reporting

no comment

Experimental design

no comment

Validity of the findings

no comment

Additional comments

All my concerns and questions have been addressed, and I recommend the acceptance of the paper。

Reviewer 2 ·

Basic reporting

no comment

Experimental design

no comment

Validity of the findings

no comment

Additional comments

Authors provide a manuscript presenting their works about methylation site prediction for proteins. This is a well written manuscript, the authors tried to use deep learning method solve the problem, the proposed deep network architecture seems working well on the prediction, achieved a better accuracy than peer methods. It would be helpful to improve the following researches of the functional and conformational changes of a protein. However, there still have a few suggestions:

1, As we all know, the position-specific scoring matrix (PSSM) is widely used as an effective representation of protein feature for a variety of protein-related problems. The authors should explain why this feature is not used.

2. The authors have to create a web server or submit all the source codes to github for reproducing the results.

3. The LSTM layer often followed by self-attention mechanism,I think this can help to improve the predictive performance.

4. The draft needs proofread.

---

## Round 0.3 · accepted · Accept

This is a well-written manuscript, the authors tried to use deep learning method to solve the problem, the proposed deep network architecture seems to work well on the prediction, achieved a better accuracy than the peer method.